# Cross-lingual Language Model Pretraining

**Alexis Conneau**[*]
Facebook AI Research
Université Le Mans
aconneau@fb.com

**Guillaume Lample**[*]
Facebook AI Research
Sorbonne Universités
glample@fb.com

## Abstract

Recent studies have demonstrated the efficiency of generative pretraining for English natural language understanding. In this work, we extend this approach to multiple languages and show the effectiveness of cross-lingual pretraining. We propose two methods to learn cross-lingual language models (XLMs): one unsupervised that only relies on monolingual data, and one supervised that leverages parallel data with a new cross-lingual language model objective. We obtain state-of-the-art results on cross-lingual classification, unsupervised and supervised machine translation. On XNLI, our approach pushes the state of the art by an absolute gain of 4.9% accuracy. On unsupervised machine translation, we obtain 34.3 BLEU on WMT'16 German-English, improving the previous state of the art by more than 9 BLEU. On supervised machine translation, we obtain a new state of the art of 38.5 BLEU on WMT'16 Romanian-English, outperforming the previous best approach by more than 4 BLEU. Our code and pretrained models are publicly available[1].

## 1 Introduction

Generative pretraining of sentence encoders [30, 20, 14] has led to strong improvements on numerous natural language understanding benchmarks [40]. In this context, a Transformer [38] language model is learned on a large unsupervised text corpus, and then fine-tuned on natural language understanding (NLU) tasks such as classification [35] or natural language inference [7, 42]. Although there has been a surge of interest in learning general-purpose sentence representations, research in that area has been essentially monolingual, and largely focused around English benchmarks [10, 40]. Recent developments in learning and evaluating cross-lingual sentence representations in many languages [12] aim at mitigating the English-centric bias and suggest that it is possible to build universal cross-lingual encoders that can encode any sentence into a shared embedding space.

In this work, we demonstrate the effectiveness of cross-lingual language model pretraining on multiple cross-lingual understanding (XLU) benchmarks. Precisely, we make the following contributions:

1. We introduce a new unsupervised method for learning cross-lingual representations using cross-lingual language modeling and investigate two monolingual pretraining objectives.

2. We introduce a new supervised learning objective that improves cross-lingual pretraining when parallel data is available.

3. We significantly outperform the previous state of the art on cross-lingual classification, unsupervised machine translation and supervised machine translation.

4. We show that cross-lingual language models can provide significant improvements on the perplexity of low-resource languages.

5. We make our code and pretrained models publicly available[1].

---

[*] Equal contribution.
[1]https://github.com/facebookresearch/XLM

## 2 Related Work

Our work builds on top of Radford et al. [30], Howard and Ruder [20], Devlin et al. [14] who investigate language modeling for pretraining Transformer encoders. Their approaches lead to drastic improvements on several classification tasks from the GLUE benchmark [40]. Ramachandran et al. [31] show that language modeling pretraining can also provide significant improvements on machine translation tasks, even for high-resource language pairs such as English-German where there exists a significant amount of parallel data. Concurrent to our work, results on cross-lingual classification using a cross-lingual language modeling approach were showcased on the BERT repository . We compare those results to our approach in Section 5.

Aligning distributions of text representations has a long tradition, starting from word embeddings alignment and the work of Mikolov et al. [27] that leverages small dictionaries to align word representations from different languages. A series of follow-up studies show that cross-lingual representations can be used to improve the quality of monolingual representations [16], that orthogonal transformations are sufficient to align these word distributions [43], and that all these techniques can be applied to an arbitrary number of languages [2]. Following this line of work, the need for cross-lingual supervision was further reduced [34] until it was completely removed [11]. We take these ideas one step further by aligning distributions of sentences and also reducing the need for parallel data.

There is a large body of work on aligning sentence representations from multiple languages. By using parallel data, Hermann and Blunsom [18], Conneau et al. [12], Eriguchi et al. [15] investigated zero-shot cross-lingual sentence classification. But the most successful recent approach of cross-lingual encoders is probably the one of Johnson et al. [21] for multilingual machine translation. They show that a single sequence-to-sequence model can be used to perform machine translation for many language pairs, by using a single shared LSTM encoder and decoder. Their multilingual model outperformed the state of the art on low-resource language pairs, and enabled zero-shot translation. Following this approach, Artetxe and Schwenk [4] show that the resulting encoder can be used to produce cross-lingual sentence embeddings. By leveraging more than 200 million parallel sentences, they obtain a new state of the art on the XNLI cross-lingual classification benchmark [12]. While these methods require a significant amount of parallel data, recent work in unsupervised machine translation show that sentence representations can be aligned in a completely unsupervised way [25, 5]. For instance, Lample et al. [26] obtained 25.2 BLEU on WMT'16 German-English without using parallel sentences. Similarly, we show that we can align distributions of sentences in a completely unsupervised way, and that our cross-lingual models can be used for a broad set of natural language understanding tasks, including machine translation.

The most similar work to ours is probably the one of Wada and Iwata [39], where the authors train a LSTM [19] language model with sentences from different languages to align word embeddings in an unsupervised way.

## 3 Cross-lingual language models

In this section, we present the three language modeling objectives we consider throughout this work. Two of them only require monolingual data (unsupervised), while the third one requires parallel sentences (supervised). We consider $N$ languages. Unless stated otherwise, we suppose that we have $N$ monolingual corpora $\{C_i\}_{i=1\ldots N}$, and we denote by $n_i$ the number of sentences in $C_i$.

### 3.1 Shared sub-word vocabulary

In all our experiments we process all languages with the same shared vocabulary created through Byte Pair Encoding (BPE) [32]. As shown in Lample et al. [25], this greatly improves the alignment of embedding spaces across languages that share either the same alphabet or anchor tokens such as digits [34] or proper nouns. We learn the BPE splits on the concatenation of sentences sampled randomly from the monolingual corpora. Sentences are sampled according to a multinomial distribution with probabilities $\{q_i\}_{i=1\ldots N}$, where: $q_i = \frac{p_i^\alpha}{\sum_{j=1}^N p_j^\alpha}$ with $p_i = \frac{n_i}{\sum_{k=1}^N n_k}$. We consider $\alpha = 0.5$. Sampling with this distribution increases the number of tokens associated to low-resource languages

and alleviates the bias towards high-resource languages. In particular, this prevents words of low-resource languages from being split at the character level.

## 3.2 Causal Language Modeling (CLM)

Our causal language modeling (CLM) task consists of a Transformer language model trained to model the probability of a word given the previous words in a sentence $P(w_t|w_1, \ldots, w_{t-1}, \theta)$. While recurrent neural networks obtain state-of-the-art performance on language modeling benchmarks [22], Transformer models are also very competitive [13]. In the case of LSTM language models, back-propagation through time [41] (BPTT) is performed by providing the LSTM with the last hidden state of the previous iteration. In the case of Transformers, previous hidden states can be passed to the current batch [1] to provide context to the first words in the batch. However, this technique does not scale to the cross-lingual setting, so we just leave the first words in each batch without context for simplicity.

## 3.3 Masked Language Modeling (MLM)

We also consider the masked language modeling (MLM) objective of Devlin et al. [14], also known as the Cloze task [36]. Following Devlin et al. [14], we sample randomly 15% of the BPE tokens from the text streams, replace them by a [MASK] token 80% of the time, by a random token 10% of the time, and we keep them unchanged 10% of the time. Differences between our approach and the MLM of Devlin et al. [14] include the use of text streams of an arbitrary number of sentences (truncated at 256 tokens) instead of pairs of sentences. To counter the imbalance between rare and frequent tokens (e.g. punctuations or stop words), we also subsample the frequent outputs using an approach similar to Mikolov et al. [28]: tokens in a text stream are sampled according to a multinomial distribution, whose weights are proportional to the square root of their invert frequencies. Our MLM objective is illustrated in Figure 1.

## 3.4 Translation Language Modeling (TLM)

Both the CLM and MLM objectives are unsupervised and only require monolingual data. However, these objectives cannot be used to leverage parallel data when it is available. We introduce a new translation language modeling (TLM) objective for improving cross-lingual pretraining. Our TLM objective is an extension of MLM, where instead of considering monolingual text streams, we concatenate parallel sentences as illustrated in Figure 1. We randomly mask words in both the source and target sentences. To predict a word masked in an English sentence, the model can either attend to surrounding English words or to the French translation, encouraging the model to align the English and French representations. In particular, the model can leverage the French context if the English one is not sufficient to infer the masked English words. To facilitate the alignment, we also reset the positions of target sentences.

## 3.5 Cross-lingual Language Models

In this work, we consider cross-lingual language model pretraining with either CLM, MLM, or MLM used in combination with TLM. For the CLM and MLM objectives, we train the model with batches of 64 streams of continuous sentences composed of 256 tokens. At each iteration, a batch is composed of sentences coming from the same language, which is sampled from the distribution $\{q_i\}_{i=1...N}$ above, with $\alpha = 0.7$. When TLM is used in combination with MLM, we alternate between these two objectives, and sample the language pairs with a similar approach.

# 4 Cross-lingual language model pretraining

In this section, we explain how cross-lingual language models can be used to obtain:

- a better initialization of sentence encoders for zero-shot cross-lingual classification
- a better initialization of supervised and unsupervised neural machine translation systems
- language models for low-resource languages
- unsupervised cross-lingual word embeddings

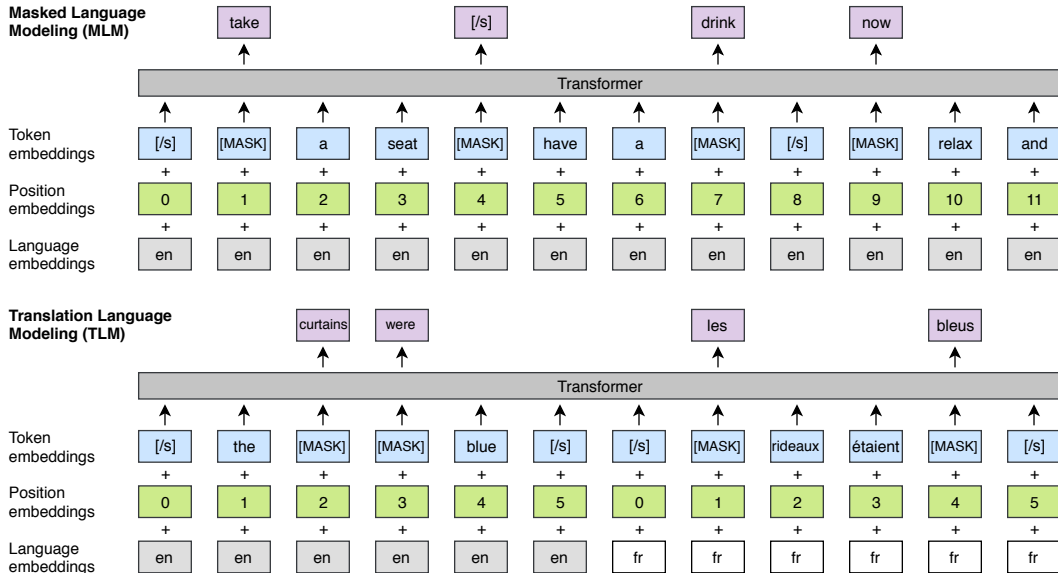

Figure 1: **Cross-lingual language model pretraining.** The MLM objective is similar to the one of Devlin et al. [14], but with continuous streams of text as opposed to sentence pairs. The TLM objective extends MLM to pairs of parallel sentences. To predict a masked English word, the model can attend to both the English sentence and its French translation, and is encouraged to align English and French representations. Position embeddings of the target sentence are reset to facilitate the alignment.

## 4.1 Cross-lingual classification

Our pretrained XLM models provide general-purpose cross-lingual text representations. Similar to monolingual language model fine-tuning [30, 14] on English classification tasks, we fine-tune XLMs on a cross-lingual classification benchmark. We use the cross-lingual natural language inference (XNLI) dataset to evaluate our approach. Precisely, we add a linear classifier on top of the first hidden state of the pretrained Transformer, and fine-tune all parameters on the English NLI training dataset. We then evaluate the capacity of our model to make correct NLI predictions in the 15 XNLI languages. Following Conneau et al. [12], we also include machine translation baselines of train and test sets. We report our results in Table 1.

## 4.2 Unsupervised Machine Translation

Pretraining is a key ingredient of unsupervised neural machine translation (UNMT) [25, 5]. Lample et al. [26] show that the quality of pretrained cross-lingual word embeddings used to initialize the lookup table has a significant impact on the performance of an unsupervised machine translation model. We propose to take this idea one step further by pretraining the entire encoder and decoder with a cross-lingual language model to bootstrap the iterative process of UNMT. We explore various initialization schemes and evaluate their impact on several standard machine translation benchmarks, including WMT'14 English-French, WMT'16 English-German and WMT'16 English-Romanian. Results are presented in Table 2.

## 4.3 Supervised Machine Translation

We also investigate the impact of cross-lingual language modeling pretraining for supervised machine translation, and extend the approach of Ramachandran et al. [31] to multilingual NMT [21]. We evaluate the impact of both CLM and MLM pretraining on WMT'16 Romanian-English, and present results in Table 3.

### 4.4 Low-resource language modeling

For low-resource languages, it is often beneficial to leverage data in similar but higher-resource languages, especially when they share a significant fraction of their vocabularies. For instance, there are about 100k sentences written in Nepali on Wikipedia, and about 6 times more in Hindi. These two languages also have more than 80% of their tokens in common in a shared BPE vocabulary of 100k subword units. We provide in Table 4 a comparison in perplexity between a Nepali language model and a cross-lingual language model trained in Nepali but enriched with different combinations of Hindi and English data.

### 4.5 Unsupervised cross-lingual word embeddings

Conneau et al. [11] showed how to perform unsupervised word translation by aligning monolingual word embedding spaces with adversarial training (MUSE). Lample et al. [25] showed that using a shared vocabulary between two languages and then applying fastText [6] on the concatenation of their monolingual corpora also directly provides high-quality cross-lingual word embeddings (Concat) for languages that share a common alphabet. In this work, we also use a shared vocabulary but our word embeddings are obtained via the lookup table of our cross-lingual language model (XLM). In Section 5, we compare these three approaches on three different metrics: cosine similarity, L2 distance and cross-lingual word similarity.

## 5  Experiments and results

In this section, we empirically demonstrate the strong impact of cross-lingual language model pretraining on several benchmarks, and compare our approach to the current state of the art.

### 5.1  Training details

In all experiments, we use a Transformer architecture with 1024 hidden units, 8 heads, GELU activations [17], a dropout rate of 0.1 and learned positional embeddings. We train our models with the Adam optimizer [23], a linear warm-up [38] and learning rates varying from $10^{-4}$ to $5.10^{-4}$.

For the CLM and MLM objectives, we use streams of 256 tokens and a mini-batches of size 64. Unlike Devlin et al. [14], a sequence in a mini-batch can contain more than two consecutive sentences, as explained in Section 3.2. For the TLM objective, we sample mini-batches of 4000 tokens composed of sentences with similar lengths. We use the averaged perplexity over languages as a stopping criterion for training. For machine translation, we only use 6 layers, and we create mini-batches of 2000 tokens.

When fine-tuning on XNLI, we use mini-batches of size 8 or 16, and we clip the sentence length to 256 words. We use 80k BPE splits and a vocabulary of 95k and train a 12-layer model on the Wikipedias of the XNLI languages. We sample the learning rate of the Adam optimizer with values from $5.10^{-4}$ to $2.10^{-4}$, and use small evaluation epochs of 20000 random samples. We use the first hidden state of the last layer of the transformer as input to the randomly initialized final linear classifier, and fine-tune all parameters. In our experiments, using either max-pooling or mean-pooling over the last layer did not work better than using the first hidden state.

We implement all our models in PyTorch [29], and train them on 64 Volta GPUs for the language modeling tasks, and 8 GPUs for the MT tasks. We use float16 operations to speed up training and to reduce the memory usage of our models.

### 5.2  Data preprocessing

We use *WikiExtractor* to extract raw sentences from Wikipedia dumps and use them as monolingual data for the CLM and MLM objectives. For the TLM objective, we only use parallel data that involves English, similar to Conneau et al. [12]. Precisely, we use MultiUN [44] for French, Spanish, Russian, Arabic and Chinese, and the IIT Bombay corpus [3] for Hindi. We extract the following corpora from the OPUS website Tiedemann [37]: the EUbookshop corpus for German, Greek and Bulgarian, OpenSubtitles 2018 for Turkish, Vietnamese and Thai, Tanzil for both Urdu and Swahili and GlobalVoices for Swahili. For Chinese and Thai we respectively use the tokenizer of Chang

| | en | fr | es | de | el | bg | ru | tr | ar | vi | th | zh | hi | sw | ur | Δ |
|---|---|---|---|---|---|---|---|---|---|---|---|---|---|---|---|---|
| *Machine translation baselines (TRANSLATE-TRAIN)* | | | | | | | | | | | | | | | | |
| Devlin et al. [14] | 81.9 | - | 77.8 | 75.9 | - | - | - | - | 70.7 | - | - | 76.6 | - | - | 61.6 | - |
| XLM (MLM+TLM) | 85.0 | 80.2 | 80.8 | 80.3 | 78.1 | 79.3 | 78.1 | 74.7 | 76.5 | 76.6 | 75.5 | 78.6 | 72.3 | 70.9 | 63.2 | 76.7 |
| *Machine translation baselines (TRANSLATE-TEST)* | | | | | | | | | | | | | | | | |
| Devlin et al. [14] | 81.4 | - | 74.9 | 74.4 | - | - | - | - | 70.4 | - | - | 70.1 | - | - | 62.1 | - |
| XLM (MLM+TLM) | 85.0 | 79.0 | 79.5 | 78.1 | 77.8 | 77.6 | 75.5 | 73.7 | 73.7 | 70.8 | 70.4 | 73.6 | 69.0 | 64.7 | 65.1 | 74.2 |
| *Evaluation of cross-lingual sentence encoders* | | | | | | | | | | | | | | | | |
| Conneau et al. [12] | 73.7 | 67.7 | 68.7 | 67.7 | 68.9 | 67.9 | 65.4 | 64.2 | 64.8 | 66.4 | 64.1 | 65.8 | 64.1 | 55.7 | 58.4 | 65.6 |
| Devlin et al. [14] | 81.4 | - | 74.3 | 70.5 | - | - | - | - | 62.1 | - | - | 63.8 | - | - | 58.3 | - |
| Artetxe and Schwenk [4] | 73.9 | 71.9 | 72.9 | 72.6 | 73.1 | 74.2 | 71.5 | 69.7 | 71.4 | 72.0 | 69.2 | 71.4 | 65.5 | 62.2 | 61.0 | 70.2 |
| XLM (MLM) | 83.2 | 76.5 | 76.3 | 74.2 | 73.1 | 74.0 | 73.1 | 67.8 | 68.5 | 71.2 | 69.2 | 71.9 | 65.7 | 64.6 | 63.4 | 71.5 |
| XLM (MLM+TLM) | **85.0** | **78.7** | **78.9** | **77.8** | **76.6** | **77.4** | **75.3** | **72.5** | **73.1** | **76.1** | **73.2** | **76.5** | **69.6** | **68.4** | **67.3** | **75.1** |

Table 1: **Results on cross-lingual classification accuracy.** Test accuracy on the 15 XNLI languages. We report results for machine translation baselines and zero-shot classification approaches based on cross-lingual sentence encoders. XLM (MLM) corresponds to our unsupervised approach trained only on monolingual corpora, and XLM (MLM+TLM) corresponds to our supervised method that leverages both monolingual and parallel data through the TLM objective. Δ corresponds to the average accuracy.

et al. [9], and the *PyThaiNLP* tokenizer. For all other languages, we use the tokenizer provided by Moses [24], falling back on the default English tokenizer when necessary. We use fastBPE to learn BPE codes and split words into subword units. The BPE codes are learned on the concatenation of sentences sampled from all languages, following the method presented in Section 3.1.

## 5.3   Results and analysis

In this section, we demonstrate the effectiveness of cross-lingual language model pretraining. Our approach significantly outperforms the previous state of the art on cross-lingual classification, unsupervised and supervised machine translation.

**Cross-lingual classification**   In Table 1, we evaluate two types of pretrained cross-lingual encoders: an unsupervised cross-lingual language model that uses the MLM objective on monolingual corpora only; and a supervised cross-lingual language model that combines both the MLM and the TLM loss using additional parallel data. Following Conneau et al. [12], we include two machine translation baselines: TRANSLATE-TRAIN, where the English MultiNLI training set is machine translated into each XNLI language, and TRANSLATE-TEST where every dev and test set of XNLI is translated to English. We report the XNLI baselines of Conneau et al. [12], the multilingual BERT approach of Devlin et al. [14] and the recent work of Artetxe and Schwenk [4].

Our fully unsupervised MLM method sets a new state of the art on zero-shot cross-lingual classification and significantly outperforms the supervised approach of Artetxe and Schwenk [4] which uses 223 million of parallel sentences. Precisely, MLM obtains 71.5% accuracy on average (Δ), while they obtained 70.2% accuracy. By leveraging parallel data through the TLM objective (MLM+TLM), we get a significant boost in performance of 3.6% accuracy, improving even further the state of the art to 75.1%. On the Swahili and Urdu low-resource languages, we outperform the previous state of the art by 6.2% and 6.3% respectively. Using TLM in addition to MLM also improves English accuracy from 83.2% to 85% accuracy, outperforming Artetxe and Schwenk [4] and Devlin et al. [14] by 11.1% and 3.6% accuracy respectively.

When fine-tuned on the training set of each XNLI language (TRANSLATE-TRAIN), our supervised model outperforms our zero-shot approach by 1.6%, reaching an absolute state of the art of 76.7% average accuracy. This result demonstrates the consistency of our approach and shows that XLMs can be fine-tuned on any language with strong performance. Similar to multilingual BERT [14], we observe that TRANSLATE-TRAIN outperforms TRANSLATE-TEST by 2.5% average accuracy, and additionally that our zero-shot approach outperforms TRANSLATE-TEST by 0.9%.

**Unsupervised machine translation**   For the unsupervised machine translation task we consider 3 language pairs: English-French, English-German, and English-Romanian. Our setting is identical to the one of Lample et al. [26], except for the initialization step where we use cross-lingual language modeling to pretrain the full model as opposed to only the lookup table.

|  |  | en-fr | fr-en | en-de | de-en | en-ro | ro-en |
|---|---|---|---|---|---|---|---|
| *Previous state-of-the-art - Lample et al. [26]* | | | | | | | |
| NMT | | 25.1 | 24.2 | 17.2 | 21.0 | 21.2 | 19.4 |
| PBSMT | | 28.1 | 27.2 | 17.8 | 22.7 | 21.3 | 23.0 |
| PBSMT + NMT | | 27.6 | 27.7 | 20.2 | 25.2 | 25.1 | 23.9 |
| *Our results for different encoder and decoder initializations* | | | | | | | |
| - | - | 13.0 | 15.8 | 6.7 | 15.3 | 18.9 | 18.3 |
| EMB | EMB | 29.4 | 29.4 | 21.3 | 27.3 | 27.5 | 26.6 |
| CLM | CLM | 30.4 | 30.0 | 22.7 | 30.5 | 29.0 | 27.8 |
| MLM | MLM | **33.4** | **33.3** | **26.4** | **34.3** | **33.3** | **31.8** |
| CLM | - | 28.7 | 28.2 | 24.4 | 30.3 | 29.2 | 28.0 |
| MLM | - | 31.6 | 32.1 | **27.0** | 33.2 | 31.8 | 30.5 |
| - | CLM | 25.3 | 26.4 | 19.2 | 26.0 | 25.7 | 24.6 |
| - | MLM | 29.2 | 29.1 | 21.6 | 28.6 | 28.2 | 27.3 |
| CLM | MLM | 32.3 | 31.6 | 24.3 | 32.5 | 31.6 | 29.8 |
| MLM | CLM | **33.4** | 32.3 | 24.9 | 32.9 | 31.7 | 30.4 |

Table 2: **Results on unsupervised MT.** BLEU scores on WMT'14 English-French, WMT'16 German-English and WMT'16 Romanian-English. For our results, the first two columns indicate the model used to pretrain the encoder and the decoder. " - " means the model was randomly initialized. EMB corresponds to pretraining the lookup table with cross-lingual embeddings, CLM and MLM correspond to pretraining with models trained on the CLM or MLM objectives.

For both the encoder and the decoder, we consider different possible initializations: CLM pretraining, MLM pretraining, or random initialization. We then follow Lample et al. [26] and train the model with a denoising auto-encoding loss along with an online back-translation loss. Results are reported in Table 2. We compare our approach with the ones of Lample et al. [26]. For each language pair, we observe significant improvements over the previous state of the art. We re-implemented the NMT approach of Lample et al. [26] (EMB), and obtained better results than reported in their paper. We expect that this is due to our multi-GPU implementation which uses significantly larger batches. In German-English, our best model outperforms the previous unsupervised approach by more than 9.1 BLEU, and 13.3 BLEU if we only consider neural unsupervised approaches. Compared to pretraining only the lookup table (EMB), pretraining both the encoder and decoder with MLM leads to consistent significant improvements of up to 7 BLEU on German-English. We also observe that the MLM objective pretraining consistently outperforms the CLM one, going from 30.4 to 33.4 BLEU on English-French, and from 28.0 to 31.8 on Romanian-English. These results are consistent with the ones of Devlin et al. [14] who observed a better generalization on NLU tasks when training on the MLM objective compared to CLM. We also observe that the encoder is the most important element to pretrain: when compared to pretraining both the encoder and the decoder, pretraining only the decoder leads to a significant drop in performance, while pretraining only the encoder only has a small impact on the final BLEU score.

**Supervised machine translation** In Table 3 we report the performance on Romanian-English WMT'16 for different supervised training configurations: mono-directional (ro→en), bidirectional (ro↔en, a multi-NMT model trained on both en→ro and ro→en) and bidirectional with back-translation (ro↔en + BT). Models with back-translation are trained with the same monolingual data as language models used for pretraining. As in the unsupervised setting, we observe that pretraining provides a significant boost in BLEU score for each configuration, and that pretraining with the MLM objective leads to the best performance. Also, while models with back-translation have access to the same amount of monolingual data as the pretrained models, they are not able to generalize as well on the evaluation sets. Our bidirectional model trained with back-translation obtains the best performance and reaches 38.5 BLEU, outperforming the previous SOTA of Sennrich et al. [33] (based on back-translation and ensemble models) by more than 4 BLEU. Similar to English-Romanian, we obtained a 1.5 BLEU improvement for English-German WMT'16 using MLM

pretraining. For English-French WMT'14 which contains significantly more supervised training data, we only obtained a minor improvement of 0.1 BLEU, which tends to indicate that the gains coming from pretraining are not as important for very high-resource settings than they are for lower-resource languages. However, in all cases we observed that convergence with pretraining is extremely fast. Typically, even for English-French, we observed that the model only needs a few epochs to converge.

| Pretraining | - | CLM | MLM |
|---|---|---|---|
| Sennrich et al. | 33.9 | - | - |
| ro $\rightarrow$ en | 28.4 | 31.5 | 35.3 |
| ro $\leftrightarrow$ en | 28.5 | 31.5 | 35.6 |
| ro $\leftrightarrow$ en + BT | 34.4 | 37.0 | **38.5** |

Table 3: **Results on supervised MT.** BLEU scores on WMT'16 Romanian-English. The previous state-of-the-art of Sennrich et al. [33] uses both back-translation and an ensemble model. ro $\leftrightarrow$ en corresponds to models trained on both directions.

**Low-resource language model** In Table 4, we investigate the impact of cross-lingual language modeling for improving the perplexity of a Nepali language model. To do so, we train a Nepali language model on Wikipedia, together with additional data from either English or Hindi. While Nepali and English are distant languages, Nepali and Hindi are similar as they share the same Devanagari script and have a common Sanskrit ancestor. When using English data, we reduce the perplexity on the Nepali language model by 17.1 points, going from 157.2 for Nepali-only language modeling to 140.1 when using English. Using additional data from Hindi, we get a much larger perplexity reduction of 41.6. Finally, by leveraging data from both English and Hindi, we reduce the perplexity even more to 109.3 on Nepali. The gains in perplexity from cross-lingual language modeling can be partly explained by the n-grams anchor points that are shared across languages, for instance in Wikipedia articles. The cross-lingual language model can thus transfer the additional context provided by the Hindi or English monolingual corpora through these anchor points to improve the Nepali language model.

| Training languages | Nepali perplexity |
|---|---|
| Nepali | 157.2 |
| Nepali + English | 140.1 |
| Nepali + Hindi | 115.6 |
| Nepali + English + Hindi | **109.3** |

Table 4: **Results on language modeling.** Nepali perplexity when using additional data from a similar language (Hindi) or a distant language (English).

| | Cosine sim. | L2 dist. | SemEval'17 |
|---|---|---|---|
| MUSE | 0.38 | 5.13 | 0.65 |
| Concat | 0.36 | 4.89 | 0.52 |
| XLM | **0.55** | **2.64** | **0.69** |

Table 5: **Unsupervised cross-lingual word embeddings** Cosine similarity and L2 distance between source words and their translations. Pearson correlation on SemEval'17 cross-lingual word similarity task of Camacho-Collados et al. [8].

**Unsupervised cross-lingual word embeddings** The MUSE, Concat and XLM (MLM) methods provide unsupervised cross-lingual word embedding spaces that have different properties. In Table 5, we study those three methods using the same word vocabulary and compute the cosine similarity and L2 distance between word translation pairs from the MUSE dictionaries. We also evaluate the quality of the cosine similarity measure via the SemEval'17 cross-lingual word similarity task of Camacho-Collados et al. [8]. We observe that XLM outperforms both MUSE and Concat on cross-lingual word similarity, reaching a Pearson correlation of 0.69. Interestingly, word translation pairs are also far closer in the XLM cross-lingual word embedding space than for MUSE or Concat. Specifically, MUSE obtains 0.38 and 5.13 for cosine similarity and L2 distance while XLM gives 0.55 and 2.64 for the same metrics. Note that XLM embeddings have the particularity of being trained together with a sentence encoder which may enforce this closeness, while MUSE and Concat are based on fastText word embeddings.

# 6    Conclusion

In this work, we show for the first time the strong impact of cross-lingual language model (XLM) pretraining. We investigate two unsupervised training objectives that require only monolingual corpora: Causal Language Modeling (CLM) and Masked Language Modeling (MLM). We show that both the CLM and MLM approaches provide strong cross-lingual features that can be used for pretraining models. On unsupervised machine translation, we show that MLM pretraining is extremely effective. We reach a new state of the art of 34.3 BLEU on WMT'16 German-English, outperforming the previous best approach by more than 9 BLEU. Similarly, we obtain strong improvements on supervised machine translation. We reach a new state of the art on WMT'16 Romanian-English of 38.5 BLEU, which corresponds to an improvement of more than 4 BLEU points. We also demonstrate that XLMs can be used to improve the perplexity of a Nepali language model, and that it provides unsupervised cross-lingual word embeddings. Without using a single parallel sentence, our MLM model fine-tuned on XNLI already outperforms the previous supervised state of the art by 1.3% accuracy on average. Our translation language model objective (TLM) leverages parallel data to improve further the alignment of sentence representations. When used together with MLM, we show that this supervised approach beats the previous state of the art on XNLI by 4.9% accuracy on average. Our code and pretrained models are publicly available.

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
