[Reviews · NeurIPS 2019]

Reviewer 1



This paper uses three techniques for incorporating multi-lingual (rather than just mono-lingual) information for pretraining contextualised representations: (i) autoregressive language modelling objective (e.g. left-to-right or right-to-left language model), (ii) masked language modelling (similar to the BERT loss, but trained on multiple languages based on a shared BPE vocabulary), and (iii) translation language modelling (extending the BERT loss to the case where parallel data are available in more than 1 language). The methods are evaluated on four tasks: (i) cross-lingual classification (XNLI), (ii) unsupervised machine translation, (iii) supervised machine translation, and (iv) low-resourcce language modelling. Strengths: - Very substantial improvements on XNLI, unsupervised machine translation, and supervised machine translation. These results are important as they showcase the strong benefit of multi-lingual (rather than just mono-lingual) pretraining for multiple important downstream tasks, and achieve new state of the art. - The methodology and the evaluation tasks are explained in sufficient technical details, which would help facilitate reproducibility. - Fairly comprehensive review of related work in Section 2. Weaknesses: see section 5 ("improvements") below. Originality: while the methods are not particularly novel (autoregressive and masked language modelling pretraining have both been used before for ELMo and BERT; this work extends these objectives to the multi-lingual case), the performance gains on all four tasks are still very impressive. - Quality: This paper's contributions are mostly empirical. The empirical results are strong, and the methodology is sound and explained in sufficient technical details. - Clarity: The paper is well-written, makes the connections with the relevant earlier work, and includes important details that can facilitate reproducibility (e.g. the learning rate, number of layers, etc.). - Significance: The empirical results constitute a new state of the art and are important to drive progress in the field. ---------- Update after authors' response: the response clearly addressed most of my concerns. I look forward to the addition of supervised MT experiments on other languages (beyond the relatively small Romanian-English dataset) on subsequent versions of the paper. I maintain my initial assessment that this is a strong submission with impressive empirical results, which would be useful for the community. I maintain my final recommendation of "8".

Reviewer 2



This paper introduces straightforward techniques for pre-training transformer architectures on monolingual corpora from many languages, as well as on parallel corpora: standard left-history conditioning for monolingual sentences, and BERT-style cloze masking for both monolingual and parallel (jointly-modeled) sentences. These techniques are combined with fine-tuning and applied to several tasks, with consistently positive results relative to SOTA baselines: cross-lingual classification (XNLI dataset), unsupervised and supervised MT, low-resource LMs, and unsupervised multilingual word embeddings. The idea of taking advantage of data in many languages is natural and powerful. The results are extremely convincing, especially given the simplicity of the techniques considered, and the general lack of extensive tuning of hyper-parameters for different tasks. The paper is very clearly written, and for the most part does a good job with reproducible detail. The previous work section is excellent. All in all, this is a highly significant empirical milestone. Some details should be specified in greater detail: Are language tags a la Johnson et al (or any other form of explicit language information) used? For MT, please be a bit more explicit about how encoder and decoder are trained, and how the final architecture comes together (eg, randomly-initialized cross-attention?). Why only show supervised results on En/Ro? Even if results weren’t as competitive on some other language pair, this would still be interesting to calibrate and understand the limitations of pre-training. Given the apparently large effect of bigger batch sizes in table 2, it would be helpful to show the effect of this setting on all tasks. Postscript, after author response: Thanks for the clear answers. Please do follow through on your promise to add supervised results for language pairs other than En/Ro.

Reviewer 3



The actual idea of cross-lingual language modeling pretraining is a natural next step in the landscape of language modeling pretraining, and coming up with the actual idea was definitely not so difficult. The difficult part was executing this very intuitive idea into solid work and empirically validate its potential. The paper is therefore not very original and methodologically novel, but it does a very solid engineering work overall. The whole idea is largely inspired by the recent BERT model (which the authors consider concurrent work, at least the multilingual version of BERT), but it is clear that they borrow the crucial idea of Masked Language Model (MLM) from the BERT paper. The only real methodological contribution - the TLM version of the MLM objective is a simple extension of MLM to cross-lingual settings, and similar extensions have been observed before in the literature on static cross-lingual word embedding learning (e.g., see the work of Gouws et al. or Coulmance et al., all from 2015). The whole methodological contribution and novelty are quite thin. Despite its low novelty, the paper is well-written and very easy follow, and all the necessary details required to reproduce the results are provided in the paper. The results are strong, although I'm a bit puzzled, given the similarity of multilingual BERT and this work, why more extensive comparisons to multilingual BERT have not been made in other tasks such as unsupervised NMT initialisation or language modeling transfer. Having said that, I would like to see more discussions on why the proposed method, being so similar to multilingual BERT, obtains large improvements over multilingual BERT on XNLI. What exactly is the main cause of difference? Is it the subwords (WordPiece vs BPE), is it the slight adaptation of the MLM objective? Is it something else? Overall, why the paper does a good job in reporting a plethora of quantitative results, I feel that deeper insights into why it works, when it works, and why it works better than some very relevant baselines (i.e., multilingual BERT or LASER) are missing from the paper. I would suggest the authors to maybe remove the parts on LM transfer and cross-lingual word embeddings, as these results do not contribute much to the paper and are very much expected and intuitive (and also stay at a very shallow level, e.g., why don't the authors report BLI scores with xling embeddings or why don't they do more LM transfer)? Removing these experiments would provide space to add more analyses of the main results. Another experiment I would like to see is adding more distant language pairs (it seems that the method works well for some more distant language pairs), but all the other language pairs tested are from the same family, and it is still debatable how well the method lends itself to applications with distant language pairs. Also, for UNMT only one method is tried. I would be interested to see if the same initialisation could be applied to some very recent robust UNMT techniques such as the method of Artetxe et al. (ACL-19). Overall, I find this paper as a good implementation of a simple idea with some strong results, but the paper requires more in-depth analyses, and the novelty of the paper remains limited (as it heavily relies on prior work, i.e., BERT). [After the author response] I thank the authors for a very insightful response which has strengthen my belief that the paper should be accepted despite some of my criticism.

[Author Response · NeurIPS 2019]

We thank the reviewers for their thorough and insightful reviews. We first respond to a common question regarding the impact of pretraining in the supervised MT setting, then provide individual answers:

**Supervised MT pretraining**  Reviewers 1 and 2 asked about the impact of pretraining in supervised MT on other language pairs than English-Romanian. We recently ran similar experiments on English-German and English-French. We observed that on En-De ($\approx$6M parallel sentences), pretraining improved the performance by 1 / 1.5 BLEU. On En-Fr ($\approx$40M parallel sentences), however, pretraining did not help. On En-Ro ($\approx$600k parallel sentences) the improvement was of 7 BLEU points. This suggests that pretraining is the most effective in low-resource scenarios. Also, in all settings (En-Fr, En-De, En-Ro) we observed that the convergence when starting from a pretrained model is extremely fast (less than 1 epoch to reach the final performance on En-Fr). However, the overall training time is longer if we include the pretraining time. We will add these results to the paper.

**Reviewer 1**

- For all language modeling experiments, we used the same architecture as for the NMT experiments: 6 layers, dimension 1024, 8 heads. For each configuration (Ne, Ne+En, Ne+Hi and Ne+Hi+En), we independently tuned the dropout, attention dropout, and optimizer learning rate. Overall, each configuration was given the same amount of hyper-parameters fine-tuning. We will detail this in the paper.

- The missing citation to ELMo was clearly an oversight on our part. We will add it to the paper.

- XNLI only provides a ground-truth training set in English, so we fine-tune on this English training set and evaluate on other languages at test time. In the "translate-train" baseline, however, we fine-tune on each of the translated training sets.

- Transformers are indeed SOTA on language modeling. We will clarify this in the paper.

**Reviewer 2**

- We use language tags to specify the current language (so if the encoder or decoder receives a German sentence, a German embedding <DE> is added at each time step). Unlike Johnson et al., we do not need to add a <TO_EN> token to the encoder input since the decoder already receives an embedding indicating the target language. Note that adding the language embedding to the encoder is actually optional, but it allows us to initialize the encoder and the decoder with a same pretrained model trained with the language embedding.

- Since we use a language model to pretrain the encoder and the decoder, we cannot pretrain the source-attention parameters that are specific to the decoder. As a result, we simply let these parameters randomly initialized (we use the default He initialization of linear layers).

**Reviewer 3**

- We tried 3 different strategies to leverage parallel data: align hidden representations of parallel sentences (as in the XNLI paper), predict whether pairs of sentences are mutual translations of each other, and the TLM objective presented in the paper. Overall, the TLM approach gave the best results, but we agree that comparing with other strategies is interesting, and we will develop this in the updated version of the paper.

- The approach of Artetxe et al. would probably benefit from cross-lingual pretraining. Indeed, their method relies on an unsupervised PBSMT training to provide back-parallel sentences, and the NMT model used in a second part could clearly be pretrained and we expect that it would improve the results even further. However, this pipeline requires a significant engineering effort (especially for the PBSMT part) and we did not try it as it is slightly beyond the scope of this paper.

- We agree that in the UNMT setting, results on distant language pairs would also be interesting. However, the majority of existing studies in UNMT focused on En-Fr, En-De and En-Ro, so we also used these language pairs for easier comparison with existing approaches.

- We believe that there are a few non-optimal strategies in the m-BERT approach that may explain the differences. First, we observed that the "next sentence prediction task" was hurting performance, which is in line with other recent studies that focused on the monolingual setting. Also, we create batches of continuous sentence streams, which results in larger attention spans than when training on sentence pairs. Overall, our training is simpler, and we observed improvements both in the monolingual setting (on GLUE tasks), and on cross-lingual tasks.

- Our work is the first to show the importance of pretraining for generation in a multilingual MT setting, and we also demonstrate that cross-lingual pretraining at the encoder/decoder level - instead of just the token embeddings - is critical for UNMT. These contributions are in our opinion quite novel, on top of the TLM approach we introduce for cross-lingual classification. Cross-lingual pretraining is maybe a natural extension, but the fact that it provides substantial improvements was not obvious in our opinion.

[Meta-Review · NeurIPS 2019]

This paper studies the problem of cross lingual language model pretraining. Pros • An important problem is studied. • Paper is well written. • Very strong empirical results. • The methodology is solid. • Comprehensive review of related work. Cons • The proposed methods are not particularly novel. All the reviewers liked the paper.